# Effectiveness of Spironolactone in Terms of Galectin-3 Levels in Patients with Heart Failure with a Reduced Ejection Fraction in the Vietnamese Population

**DOI:** 10.3390/healthcare11020253

**Published:** 2023-01-13

**Authors:** Son Kim Tran, Toan Hoang Ngo, Tin Trung Lai, Giang Khanh Truong, Khoa Dang Dang Tran, Phuong Minh Vo, Phi The Nguyen, Phi Hoang Nguyen, Thuan Tuan Nguyen, Oanh Thi Kim Nguyen, Thang Nguyen, Kien Trung Nguyen, Hung Do Tran

**Affiliations:** 1Department of Internal Medicine, Can Tho University of Medicine and Pharmacy, Can Tho City 900000, Vietnam; 2Department of Cardiology, Can Tho Cardiovascular Hospital, Can Tho City 900000, Vietnam; 3Faculty of Medicine, Can Tho University of Medicine and Pharmacy, Can Tho City 90000, Vietnam; 4Department of Pharmacology and Clinical Pharmacy, Can Tho University of Medicine and Pharmacy, Can Tho City 900000, Vietnam; 5Faculty of Nursing and Medical Technology, Can Tho University of Medicine and Pharmacy, Can Tho City 90000, Vietnam

**Keywords:** heart failure, reduced ejection fraction, galectin-3

## Abstract

Background: Galectin-3 is a biomarker that has been demonstrated to play a significant role in myocardial fibrosis and remodeling in the pathogenesis of heart failure. Furthermore, spironolactone has the ability to control galectin-3 levels in heart failure patients. Objectives: The aim of our study was to determine the factors associated with the increase in galectin-3 and the alteration of galectin-3 concentration in patients with heart failure with a reduced ejection fraction after 12 weeks of treatment with spironolactone. Materials and methods: A cross-sectional descriptive study was conducted on 122 patients with heart failure with a reduced ejection fraction. Those patients were nonusers of spironolactone and presented for examination or had been hospitalized at the Can Tho Cardiovascular Hospital in Vietnam. The demographic and cardiovascular risk factor details were obtained at baseline, and galectin-3 levels were measured at baseline and also 12 weeks after taking spironolactone 25 mg once daily vs. 50 mg once daily. Results: The median baseline galectin-3 was 54.82 ± 26.06. Galectin-3 levels were positively correlated with age, NT-proBNP, and negatively correlated between EF and galectin-3 levels (*p* < 0.05). After 12 weeks of treatment with spironolactone, the galectin-3 concentration decreased from 54.82 ± 26.06 to 44.20 ± 24.36 (*p* < 0.05). According to the subgroup analysis, the average concentration of galectin-3 decreased the most in the group of patients with grade 3 hypertension and NYHA class III heart failure. The 50 mg once-daily dose of spironolactone significantly improved galectin-3 concentrations compared with the 25 mg once-daily group, at 17.11 ± 20.81 (*p* < 0.05) (reduced 29.05%) and 3.46 ± 6.81 ng/mL (*p* < 0.05) (reduced 6.87%), respectively. Conclusion: Treatment with spironolactone played an essential role in reducing galectin-3 concentrations, especially spironolactone 50 mg once daily, which showed a significant effect on reducing galectin-3 compared with a 25 mg once-daily dose.

## 1. Introduction

Heart failure has long been considered a serious disease with high morbidity and mortality. It is currently a global public health problem, affecting an estimated 26 million worldwide, and its prevalence is increasing, mainly in the elderly. In the United States alone, the prevalence is 5.7 million, and there are 670,000 new cases per year [1].

In this day and age, numerous biomarkers, including BNP (brain natriuretic peptide) and NT-proBNP, are used to determine the diagnosis and prognosis for HF (heart failure) patients. Recently, galectin-3 has been proposed as a novel biomarker. Galectin-3 belongs to the family of soluble lectins. It is mainly produced and secreted by activated macrophages. Galectin-3 plays a central role in promoting myofibroblast proliferation, which leads to tissue remodeling and myocardial fibrosis [2]. Galectin-3 was found to be significantly increased in patients with chronic HF, and several studies have determined its prognostic value in patients with HF for all-cause mortality [3]. Through data from clinical and pathophysiological studies, it has been shown that galectin-3 has the potential to be a novel and promising biomarker to determine cardiac fibrosis and remodeling. Moreover, it has been shown to be a potent prognostic factor in HF patients [2,3,4].

Spironolactone has long been indicated to improve symptoms and reduce hospitalizations and mortality in patients with HFrEF (heart failure with a reduced ejection fraction) [5]. The studies have indicated that plasma galectin-3 expression is stimulated by aldosterone in patients with HF, particularly HFrEF patients. As a result, it provides a new treatment approach through controlling galectin-3 concentrations to improve fibrosis and remodeling in the progression of HF [6,7,8]. However, limited data have been published on the association and interaction between spironolactone treatment and galectin-3 concentrations in patients with HF, especially HFrEF patients. To address this question, we conducted this study with two objectives: (i) to determine the factors related to the elevation of plasma galectin-3 concentration in HFrEF patients and (ii) to evaluate the effectiveness of spironolactone in terms of the modulation of plasma galectin-3 concentration.

## 2. Materials and Methods

### 2.1. Study Population

#### 2.1.1. Materials

The study was performed on 122 heart failure patients with a reduced ejection fraction who attended the Can Tho Cardiovascular Hospital from May 2018 to May 2019.

#### 2.1.2. Inclusion Criteria 

All patients were diagnosed with heart failure with a reduced ejection fraction according to the ESC criteria 2016 [9] and were given an unchanged update in 2021 [10] and agreed to participate in the study.

#### 2.1.3. Exclusion Criteria

The following were excluded:-Patients currently taking spironolactone prior to participating in this study.-Patients with an intolerance or contraindication to spironolactone: impaired renal function with an eGFR of less than 60 mL/min/1.73 m^2^, liver injury (AST and ALT levels three times higher than the normal range), serum potassium levels > 5.0 mmol/L [9,10].-Patients with a low life expectancy (<1 year).-Patients with advanced cancer (Figure 1).

### 2.2. Methods

#### 2.2.1. Study Design

A cross-sectional descriptive research method was followed (prospective study).

#### 2.2.2. Sample Size

Random sampling was used, including all patients who satisfied the sampling criteria and did not meet the exclusion criteria.

### 2.3. Data Collection

Anthropometric characteristics and risk factors such as age, sex, BMI, waist circumference, smoking status, time to heart failure detection, chief complaint, family history of early cardiovascular disease, history of coronary artery disease, blood pressure level, and NYHA index were obtained via physical examination. All patients participating in the study were treated with spironolactone for 12 weeks. Their initial dose was 25 mg once daily, and they were re-evaluated after 2 weeks. Patients who were symptomatic but for whom spironolactone was well tolerated, with no contraindications, could have their dose increased to the target dose of 50 mg once daily, according to the European Society of Cardiology’s 2021 recommendation (Class IA recommendation) in order to improve their symptoms, mortality and hospitalization rate due to heart failure [10]. The galectin-3 concentrations at baseline and at 12 weeks were used to evaluate treatment outcomes.

### 2.4. Data Analysis

A cut-off point of NT-proBNP <125 pg/mL has been proposed to rule out the diagnosis of chronic heart failure [11], with 300 pg/mL for acute heart failure, and the diagnosis was confirmed with a cut-off point of NT-proBNP >450 pg/mL for patients aged < 55 years, >900 pg/mL for those aged 55 to 75, and >1800 pg/mL for those aged > 75 years [12]. Waist circumference increases significantly when the cut-off point is >90 cm for males and >80 cm for females [13]. The normal plasma galectin-3 concentration ranges from 1.4 to 22 ng/mL, and a cut-off point of 22.1 ng/mL is commonly used for clinical diagnosis. In addition, galectin-3 levels are considered elevated when galectin-3 concentration ≥ 22.1 ng /mL [14]. 

### 2.5. Measurements

Left ventricular systolic function was measured by echocardiography using a Siemens X500 ultrasound machine at the Diagnostic Imaging Department of the Can Tho Cardiovascular Hospital, and LVEF < 40%, measured using the Simpson’s method was classified as heart failure with a reduced ejection fraction according to the 2016 ESC [9], with no change in definition according to the 2021 ESC [10]. NT-proBNP levels were measured using a Cobas 6000 analyzer at the Laboratory Department of the Can Tho Cardiovascular Hospital and standardized based on the sandwich immunoassay using an electrochemiluminescence immunoassay (ECLIA). 

Serum galectin-3 concentration was measured using an ARCHITECT machine of Abbott USA at the Can Tho Cardiovascular Hospital, which was standardized with the galectin-3 assay using chemiluminescent microparticle immunoassay (CMIA) for the quantitative determination of galectin-3 in human serum and EDTA plasma [15].

### 2.6. Statistical Analysis

The obtained data were analyzed using SPSS 20.0 software. Qualitative variables were presented as frequencies and percentages (%). Quantitative variables were presented as the mean ± standard deviation (SD), maximum value, and minimum value. The association between increased galectin-3 and risk factors was determined by logistic regression. A linear regression model between the correlation of risk factors and galectin-3 concentration was expressed as a multivariable regression equation. We used Chi-square tests (X2) to evaluate the association between two categorical variables, and where there was at least one cell with an expected value of 5, this was corrected using Fisher’s exact test. The paired-sample t test was used to compare the mean galectin-3 concentrations at two different times: at baseline and after 12 weeks of treatment. The correlation between the concentration of galectin-3 and the related factors was expressed using the Pearson correlation coefficient, where r > 0 represents a positive correlation and r < 0 represents a negative correlation. The difference was statistically significant when *p* < 0.05.

## 3. Results

### 3.1. Baseline Subject Characteristics

The research population is shown in Table 1.

### 3.2. The Proportion of Increased Galectin-3 and the Relationship between Some Cardiovascular Risk Factors and Galectin-3

When considering the Pearson correlation between pretreatment galectin-3 concentrations and risk factors, we discovered that age, time to the detection of heart failure, NT-proBNP, BMI and waist circumference were all positively correlated (*p* < 0.001), while left ventricular systolic function (EF%) was negatively correlated (*p* < 0.001) (Table 2).

Several factors, such as time to the detection of heart failure, NT-proBNP level, left ventricular systolic function (EF), and pre-treatment plasma galectin-3 concentration, showed a linear correlation (*p* < 0.05). The time to the detection of heart failure had the greatest impact on galectin-3 concentration, and the formula for linear regression (Table 3) was constructed as follows:

Galectin-3 Concentration = 34.231 + 2.265. (Time to Detection of Heart Failure) + 0.003. (NT-ProBNP Concentration)–0.992.(EF%). 

### 3.3. The Variation in Galectin-3 after 12 Weeks of Treatment with Spironolactone

The mean concentration of galectin-3 decreased the most in the group of patients with grade 3 hypertension, from 71.67 ± 42.44 to 50.19 ± 31.53. The alteration gradually decreased depending on the patient’s blood pressure level: 14.35 ± 10.64 in grade 2, 9.29 ± 23.43 in grade 1 hypertension, and the lowest at only 8.34 ± 11.67 occurred in the group without hypertension (*p* < 0.05). According to the NYHA functional classification, we discovered a noticeable decrease in the mean concentration of galectin-3 in class III heart failure, from 63.04 ± 20.79 to 50.76 ± 18.70 (*p* < 0.001) and lowered the most in the NYHA IV group, however, *p* = 0.05 due to the small sample size.

Eventually, when considering the daily dose of spironolactone, the alteration was most pronounced in the 50 mg per day spironolactone group, with galectin-3 levels of 41.78 ± 23.60 at 12 weeks of treatment, compared with 58.89 ± 23.29 at the baseline. The variation in galectin-3 concentration was 17.11 ± 20.81 (*p* < 0.001). The mean variation in galectin-3 decreased by 3.46 ± 6.81 in the 25 mg spironolactone group, and the difference was statistically significant (*p* < 0.001) (Table 4), (Figure 2).

## 4. Discussion

### 4.1. Some Factors Associated with Elevated Serum Galectin-3 Levels in Patients with Heart Failure with a Reduced Ejection Fraction

Numerous studies have also recorded the association between the above cardiovascular risk factors and galectin-3 levels in heart failure patients. Typically, the DEAL-HF study [3] showed similar results to ours, in which age, history of coronary heart disease, BMI, and NT-proBNP were also associated with an increase in galectin-3 in patients with heart failure (*p* < 0.05), and galectin-3 concentration was positively correlated with age and NT-proBNP (r = 0.318 *p* < 0.001; r = 0.265 *p* < 0.001) (Table 2). However, there was a difference between the results in terms of BMI in the DEAL-HF study and in our study. The DEAL-HF study recorded a negative correlation (r= -0.154, *p* = 0.022), while our study recorded a positive correlation with BMI (Table 2). This difference probably comes from the research object. Our study was performed in the Vietnamese population, with the mean BMI for males and females being 22.72 ± 3.87 kg/m^2^ and 23.07 ± 3.90 kg/m^2^, respectively. In contrast, the DEAL-HF study was performed in the Netherlands, where the mean BMI was 26.3 kg/m^2^. There is a certain difference in the anthropometric weight and height between these two populations, leading to the difference in BMI between the two studies. As a result, the research outcomes will probably have some differences [3]. Data from the HF-ACTION study on 895 patients with heart failure also showed quite similar results to our study in terms of the association between galectin-3 and age, history of coronary heart disease, BMI, and NT-proBNP (*p* < 0.05). Additionally, galectin-3 concentrations were positively correlated with NT-proBNP (r = 0.3 *p* < 0.001). However, there was a difference in that there was no association between EF and galectin-3 (*p* = 0.133) in the HF-ACTION study [16]. The reason for this discrepancy was probably due to the galectin-3 cut-off point. The HF-ACTION study used a cut-off point for galectin-3 of 14.0 ng/mL, whereas our study used a cut-off point of 22.1 ng/mL [16]. In a study by Rudoff A. de Boer and partners on 592 heart failure cases followed up for 18 months, there was also an association observed between galectin-3 and several factors such as age and NT-proBNP concentration (*p* < 0.05); however, no association was seen between galectin-3 and BMI, EF and history of myocardial infarction (*p* = 0.56; *p* = 0.093; *p* = 0.74) [4].

Our study and the studies cited above ascertained an association between a history of coronary heart disease and elevated galectin-3 in patients with heart failure. In fact, coronary artery disease has long been considered one of the principal risk factors for heart failure, and atherosclerosis plays a crucial role in the pathogenesis of coronary artery disease. Galectin-3 is a biomarker implicated in atherosclerosis, particularly in coronary arteries. This mechanism is explained by the fact that galectin-3 is considered to be involved in the process of plaque formation via the effects of inflammation, remodeling, and vascular fibrosis [17]. Galectin-3 has also been demonstrated to be associated with unstable atherosclerotic plaques; individuals with unstable coronary artery disease had higher galectin-3 concentrations than those with stable coronary artery disease [18]. Moreover, the studies mentioned above have also shown that there is an association between galectin-3 and arterial stiffness [19]. According to one of our studies, arterial stiffness is a crucial risk factor in hypertension, and this study has also shown that the atherogenic index is associated with left ventricular hypertrophy, which is the initial phase of ventricular remodeling in patients with heart failure [20]. NT-proBNP is a biomarker that is secreted as a result of myocyte stretching. In heart failure, especially heart failure with a reduced ejection fraction, fluid retention in the ventricles occurs almost continuously, leading to dilatation of the ventricular wall, which increases NT-proBNP secretion [21]. NT-proBNP levels are also considered to be diagnostic and prognostic factors in patients with heart failure [10,22]. Therefore, it is quite appropriate that our research discovered an association between increased concentrations of galectin-3 and NT-proBNP. 

Galectin-3-related waist circumference was also observed in the ARIC study, similar to our research [23]. Obesity, along with an increase in waist circumference, has long been considered to be a risk factor for cardiovascular disease, especially in patients with heart failure. Preclinical studies have shown that inflammation, as well as lipotoxicity, can affect cardiomyocyte metabolism, causing myocardial cell death, fibrosis, and myocardial remodeling—one phase in the process of increasing plasma galectin-3 [23,24]. In addition, galectin-3 has been implicated in metabolism and in insulin resistance [25]; in patients with heart failure, especially heart failure with a reduced ejection fraction, there was a higher rate of insulin resistance than in the group with a preserved ejection fraction [26]. Thus, we posed the intriguing question concerning whether there is a distinction between galectin-3, insulin resistance, and subgroups of patients with heart failure.

### 4.2. Multivariate Correlation between Galectin-3 and Risk Factors in Patients with Heart Failure

According to the study outcomes, we demonstrated that the time to the detection of heart failure, the concentration of NT-proBNP, and the ejection fraction had a multivariable correlation with the variation in galectin-3, in which the detection time factor had the strongest effect on galectin-3 levels. Author Jennifer E. Ho et al. analyzed the baseline data of the Framingham study and showed a correlation between increased galectin-3 and NT-proBNP levels through multivariate analysis, similar to our study [27]. Ravi V. Shah and colleagues conducted a study on 599 hospitalized heart failure patients and discovered that galectin-3 was positively correlated with age (r = 0.26, *p* = 0.006) and NT-proBNP concentration (r = 0.39, *p* < 0.001), while it was negatively correlated with glomerular filtration rate (r = −0.42, *p* < 0.001) and positively correlated with NT-proBNP concentration (r = 0.39, *p* < 0.001). Through multivariate analysis, the author also noted the correlation between galectin-3 concentration and NT-proBNP concentration along with ejection fraction (EF) [28]. Heart failure is a chronic disease; over time, heart failure will worsen, and left ventricular systolic function will gradually decrease if not appropriately treated until EF decreases to < 40% when the function of the heart is no longer guaranteed. Then, fluid retention in the cardiac chambers leads to ventricular dilatation. As a consequence, the secretion of NT-proBNP will increase [29,30]. Galectin-3 itself has been shown to be a novel inflammatory biomarker, which raises an intriguing question as to whether inflammation in patients without heart failure or hemodynamic disturbances is associated with increased NT-proBNP and galectin-3 or not [31]. Currently, there are not many studies discussing this issue.

### 4.3. Results of Controlling Serum Galectin-3 Levels Using Spironolactone in Heart Failure Patients with a Reduced Ejection Fraction

In the ALDO-DHF study, Frank Edelmann et al. analyzed 377 out of 422 patients with heart failure with a stable preserved ejection fraction who were randomized to the ALDO-DHF trial and divided into two groups: those taking spironolactone 25 mg per day and the control group. All patients in this analysis had galectin-3 levels monitored at baseline, six, and 12 months. The study results showed that galectin-3 was not related to the therapeutic effects of spironolactone, and spironolactone did not change plasma galectin-3 concentrations over time [32]. This is distinct from our study results showing that spironolactone improved galectin-3 concentrations, and this change was statistically significant (*p* < 0.01). This disparity is most likely due to the fact that our study focused on patients with a reduced ejection fraction, whereas ALDO-DHF was performed on patients with a preserved ejection fraction.

In contrast, the studies of two authors, Gucuk Ipek and Deveci, produced similar results to ours [6,7]. Specifically, in the study of Gucuk Ipek et al., performed on 14 heart failure patients with an EF ≤ 35% and NYHA class II, treated with spironolactone 25 mg per day for 3 months, it was demonstrated that galectin-3 concentration decreased significantly between pre- and post-treatment (1.49 ± 0.58 vs. 0.98 ± 0.29, *p* = 0.01). In addition, the authors also noted an association between a decrease in galectin-3 levels and improved left ventricular systolic and diastolic function in this group of patients (*p* = 0.01 and *p* = 0.02) [7]. Similarly, this alteration was also observed in the study by Deveci et al. in 112 patients with heart failure (EF < 35%) during 6 months of spironolactone treatment; the galectin-3 concentration decreased from 39 ± 21 ng/mL to 33 ± 22 ng/mL [6]. The results of these two studies were quite similar to our study, where the mean concentration of galectin-3 also decreased from 54.82 ± 26.06 to 44.20 ± 24.36 at 12 weeks post-treatment, and this change was statistically significant (*p* < 0.01). The similarity between our study and the two studies mentioned above likely comes from the fact that our study was performed on patients with heart failure with a reduced ejection fraction, and spironolactone has long been identified to improve prognoses in this patient population. SGLT2i and ARNI are considered to be effective drug classes in the treatment of HFpEF [33,34]. However, there are currently no studies evaluating their effect on reducing galectin-3 levels in this population of participants. 

In 58 patients treated with spironolactone 25 mg per day, galectin-3 decreased from 50.34 ± 28.33 ng/mL pre-treatment to 46.87 ± 25.09 ng/mL after 12 weeks of treatment, with a change of 3.46 ± 2. 6.81 ng/mL (*p* < 0.05); this result is almost similar to the study of Gucuk Ipek et al. [7] However, we observed a marked change in galectin-3 concentrations in patients who were taking 50 mg per day of spironolactone, with galectin-3 concentrations decreasing from 58.89 ± 23.29 ng/mL pre-treatment to 41.78 ± 1. 23.60 ng/mL post-treatment (*p* < 0.05). By comparing this ratio, we discovered that, after taking spironolactone, galectin-3 decreased the most (29.05%) with the dose of 50 mg per day, compared with the 25 mg per day dose (6.87%) (*p* < 0.05); this was not observed in the study by Gucuk Ipek and the ALDO-DHF study, which only included a 25 mg per day dose of spironolactone [7,32].

Many studies have shown a relationship between increased levels of galectin-3 and fibrosis remodeling in the myocardium. In the report by Lax et al., the researchers found that in a rat model of ventricular dysfunction after myocardial infarction, there was an increase in several biomarkers of fibrosis and inflammation, including galectin-3. Treatment with anti-mineralocorticoids reduced these biomarkers, including galectin-3. Therefore, the authors concluded that there was no difference in the regulation of these fibroblasts between eplerenone and spironolactone [8]. Controlling galectin-3 levels with spironolactone improved fibrosis, inflammation, and remodeling in the heart [4,8]. Moreover, in patients with heart failure, especially heart failure with a reduced ejection fraction, the reduction in galectin-3 concentrations achieved through treatment with spironolactone showed a significant improvement in cardiac function [7].

### 4.4. Strengths and Limitations

Although our study showed that the use of spironolactone at a dose of 50 mg per day provided a greater reduction in galectin-3 concentration than a dose of 25 mg in patients with heart failure with a reduced ejection fraction, certain limitations were present in our study. First, all patients in our study were taken from one center, and the follow-up period was only 3 months. Therefore, it has not been demonstrated whether long-term management of plasma galectin-3 improves mortality and hospitalization in heart failure patients, especially in those with a reduced ejection fraction. Second, whether good control of plasma galectin-3 levels resulting from medication such as SGLT2 or ARNI in patients with heart failure with a preserved ejection fraction further enhances the survival prognosis in this group of patients is still unknown. As a result, we recommend a large, multicenter, and long-term follow-up study involving the combination of spironolactone and the mainstay drug classes in heart failure treatment, such as SGLT2, ARNI, and beta-blockers, including those patients with a preserved ejection fraction, to determine whether these classes of drugs are effective in reducing galectin-3 levels.

## 5. Conclusions

In brief, numerous risk factors, including coronary heart disease history, NT-proBNP levels, EF, waist circumference, BMI, and time to diagnosis of heart failure, were associated with increased galectin-3 levels in patients with heart failure with a reduced ejection fraction. Galectin-3 concentrations were demonstrated to be significantly improved by taking spironolactone at a dose of 50 mg once daily, and this improvement was more evident than in patients treated with spironolactone 25 mg once daily. The alteration in galectin-3 concentration post-treatment was also confirmed according to the classification of blood pressure, as well as the NYHA functional classification.

## Figures and Tables

**Figure 1 healthcare-11-00253-f001:**
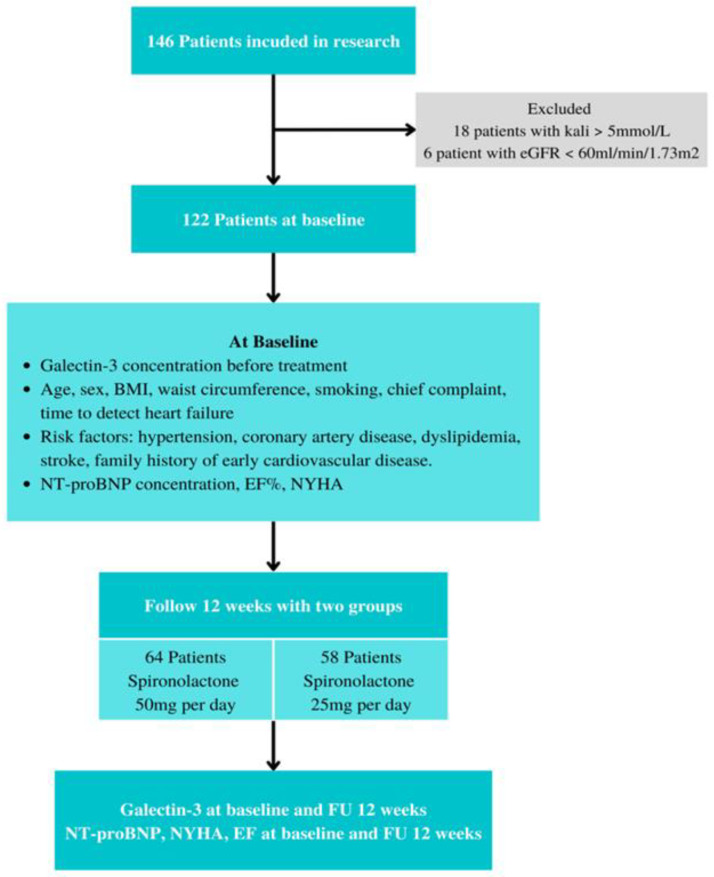
Participant flow diagram.

**Figure 2 healthcare-11-00253-f002:**
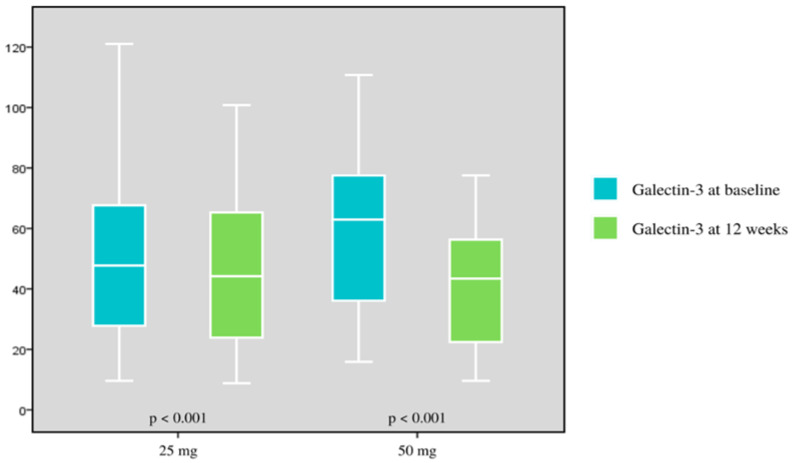
Galectin-3 concentrations before and after a 12-week treatment with two dosages of 25 mg per day and 50 per day spironolactone.

**Table 1 healthcare-11-00253-t001:** Baseline Characteristics of the Study Population.

Characteristics		N (%) or Mean ± SD
Male		54 (44.3)
Age	31–49	7 (5.74)
	50–69	73 (59.83)
	≥ 70	42 (34.43)
BMI (kg/m^2^)	Male	22.72 ± 3.87
	Female	23.07 ± 3.90
Waist circumference (cm)	Male	87.17 ± 8.90
	Femal	85.10 ± 12.45
Time to detection of heart failure (years)		5.10 ± 4.05
Coronary artery disease		45 (36.9)
Cerebrovascular accident		17 (13.9)
Dyslipidemia		107 (87.7)
Smoking		20 (16.4)
Family history of early cardiovascular disease		69 (56.6)
Hypertension		75 (61.5)
NYHA	II	44 (36.1)
	III	74 (60.6)
	IV	4 (3.3)
Hypertension	Normal	47 (38.5)
	Grade 1	44 (36.1)
	Grade 2	24 (19.7)
	Grade 3	7 (5.7)
Systolic blood pressure (mmHg)	Male	142.96 ± 22.29
	Female	140.29 ± 23.88
Diastolic blood pressure (mmHg)	Male	87.78 ± 14.46
	Female	85.53 ± 15.24
NT-proBNP	Male	4557.26 ± 2897.29
	Female	4559.18 ± 2466.73
EF%	Male	33.17 ± 5.54
	Female	32.81 ± 5.08
Spironolacton	25 mg	58 (47.5)
	50 mg	64 (52.5)
Medications	ACEIs/ARBs	75 (61.5)
	Beta-blockers	56 (45.9)
	Diuretics	28 (22.95)
	Digoxin	15 (12.30)
	Antiplatelets	62 (50.82)
	Statin	114 (93.44)
	Nitrate	8 (6.56)
Galectin-3 (ng/mL)		54.82 ± 26.06

**Table 2 healthcare-11-00253-t002:** Pearson correlation between pretreatment galectin-3 concentrations and risk factors.

Cardiovascular Risk Factors	Galectin-3	
	r	*p*
Age	0.325	<0.001
Time to detection of heart failure	0.714	<0.001
NT-proBNP	0.744	<0.001
EF	−0.701	<0.001
BMI	0.344	<0.001
Waist circumference	0.388	<0.001

**Table 3 healthcare-11-00253-t003:** Multivariate correlation between pre-treatment galectin-3 concentrations and risk factors.

Cardiovascular Risk Factors	β	Std. Error	*t*	95% CI β	*p*
Age	0.218	0.149	1.461	−0.078–0.514	0.147
Time to detection of heart failure	2.265	0.444	5.098	1.385–3.146	<0.001
Systolic blood pressure (mmHg)	−0.183	0.147	−1.245	−0.475–0.108	0.216
Diastolic blood pressure (mmHg)	0.374	0.229	1.631	−0.080–0.827	0.106
BMI	0.537	0.465	1.154	−0.385–1.459	0.251
Waist circumference	−0.064	0.166	−0.385	−0.393–0.265	0.701
NT-proBNP	0.003	0.001	3.716	0.001–0.005	<0.001
EF	−0.992	0.423	−2.345	−1.830–(−0.154)	0.021

**Table 4 healthcare-11-00253-t004:** Galectin-3 levels at baseline and at the end of the follow-up.

Risk Factors		Galectin-3	*n*	( $\bar{X}$ + SD)	Changing	*p* (Paired Sample *t*-test)
Galectin-3		Baseline	122	54.82 ± 26.06	10.62 ± 17.15	<0.001
		Follow up	122 *	44.20 ± 24.36		
Classification of blood pressure	Normal	Baseline	47	46.43 ± 23.96	8.34 ± 11.67	<0.001
		Follow up	47 *	38.09 ± 21.15		
	Grade 1	Baseline	44	58.10 ± 25.58	9.29 ± 23.43	0.012
		Follow up	44 *	48.81 ± 28.30		
	Grade 2	Baseline	24	60.34 ± 21.15	14.35 ± 10.64	<0.001
		Follow up	24 *	45.99 ± 18.38		
	Grade 3	Baseline	7	71.67 ± 42.44	21.49 ± 16.53	0.014
		Follow up	7 *	50.19 ± 31.53		
NYHA classification	NYHA II	Baseline	44	36.41 ± 21.52	6.85 ± 22.70	0.052
		Follow up	44 *	29.56 ± 24.72		
	NYHA III	Baseline	74	63.04 ± 20.79	12.28 ± 12.60	<0.001
		Follow up	74 *	50.76 ± 18.70		
	NYHA IV	Baseline	4	105.38 ± 13.52	21.48 ± 13.46	0.05
		Follow up	4 *	83.90 ± 18.91		
Spironolacton	25 mg per day	Baseline	58	50.34 ± 28.33	3.46 ± 6.81	<0.001
		Follow up	58	46.87 ± 25.09		
	50 mg per day	Baseline	64	58.89 ± 23.29	17.11 ± 20.81	<0.001
		Follow up	64	41.78 ± 23.60		

* The numeral of patients followed up from the baseline.

## Data Availability

The datasets generated and/or analyzed during the current study are available from the corresponding author on reasonable request.

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
