# Peer review of "Effectiveness of Spironolactone in Terms of Galectin-3 Levels in Patients with Heart Failure with a Reduced Ejection Fraction in the Vietnamese Population"

_healthcare, 2023, doi:10.3390/healthcare11020253_

Round 1
Reviewer 1 Report
1. Highlight the novelties of this research.
2. Please provide a clear conclusion and perspective.
3. As shown in Figure 2, the differences are not obvious. Please explain that.
4. Provide one cartoon figure to exhibit the relationship between cardiovascular risk factors and Galectin-3 levels.
5. I think it’s interesting research. If these above issues are handled properly, I would recommend it publish.
Author Response
First, the authors sincerely thank the reviewers for agreeing to read and give us valuable contributions.
Here we would like to reply to the comments of the reviewer as follows:
- Highlight the novelties of this research.
Response 1: The average concentration of Galectin-3 decreased the most in the group of patients with grade 3 hypertension and NYHA class III. The 50 mg once daily dose of spironolactone significantly improved galectin-3 concentrations over the 25 mg once daily group, which was 17.11 ± 20.81 (p < 0.05) (reduced 29.05%) and 3.46 ± 6.81ng/ml (p<0.05) (reduced 6.87%), respectively.
- Please provide a clear conclusion and perspective.
Response 2: Treatment with spironolactone plays an essential role in reducing galectin-3 concentrations, especially spironolactone 50 mg once daily, which showed a significant effect on reducing galectin-3 compared with 25 mg once daily.
- As shown in Figure 2, the differences are not obvious. Please explain that
Response 3: We wanted to use a graph to show the change before and after treatment with 2 doses of spironolactone 25mg and 50mg in galectin-3 concentrations, however, changes (not much) were available in Table 6
Table 6. Galectin-3 levels at baseline and at the end of the follow-up.
Risk factors |
Galectin-3 |
n |
Changing |
p (paired sample t-test) |
||
Galectin-3 |
Baseline |
122 |
54.82 ± 26.06 |
10.62 ± 17.15 |
<0.001 |
|
Follow up |
122 |
44.20 ± 24.36 |
||||
Classifi-cation of blood pressure
|
Normal |
Baseline |
47 |
46.43 ± 23.96 |
8.34 ± 11.67 |
<0.001 |
Follow up |
47 |
38.09 ± 21.15 |
||||
Grade 1 |
Baseline |
44 |
58.10 ± 25.58 |
9.29 ± 23.43 |
0.012 |
|
Follow up |
44 |
48.81 ± 28.30 |
||||
Grade 2 |
Baseline |
24 |
60.34 ± 21.15 |
14.35 ± 10.64 |
<0.001 |
|
Follow up |
24 |
45.99 ± 18.38 |
||||
Grade 3 |
Baseline |
7 |
71.67 ± 42.44 |
21.49 ± 16.53 |
0.014 |
|
Follow up |
7 |
50.19 ± 31.53 |
||||
NYHA classify-cation |
NYHA II |
Baseline |
44 |
36.41 ± 21.52 |
6.85 ± 22.70 |
0.052 |
Follow up |
44 |
29.56 ± 24.72 |
||||
NYHA III |
Baseline |
74 |
63.04 ± 20.79 |
12.28 ± 12.60 |
<0.001 |
|
Follow up |
74 |
50.76 ± 18.70 |
||||
NYHA IV |
Baseline |
4 |
105.38 ± 13.52 |
21.48 ± 13.46 |
0.05 |
|
Follow up |
4 |
83.90 ± 18.91 |
||||
Spironolacton |
25 mg per day |
Baseline |
58 |
50.34 ± 28.33 |
3.46 ± 6.81 |
<0.001 |
Follow up |
58 |
46.87 ± 25.09 |
||||
50 mg per day |
Baseline |
64 |
58.89 ± 23.29 |
17.11 ± 20.81 |
<0.001 |
|
Follow up |
64 |
41.78 ± 23.60 |
- Provide one cartoon figure to exhibit the relationship between cardiovascular risk factors and Galectin-3 levels.
- I think it’s interesting research. If these above issues are handled properly, I would recommend it publish.
Thank you for giving us a opportunity

Reviewer 2 Report
Please check the attachment.

Author Response
First, the authors sincerely thank the reviewers for agreeing to read and give us valuable contributions.
Here we would like to reply to the comments of the reviewer as follows:
- There are a lot of spelling and grammar mistakes in the article, and language needs extensive editing
Response 1: We utilized healthcare English editing service.
- Line 26, what’s the meaning of “rate”, percentage of patients with increased Galectin-3?
Response 3: We apologized, we wrote it wrong, there is no target percentage in this abstract. We will delete and re-edit according to the feedback.
- Line 34-35, the sentence ”The association between time to detect heart failure, NT-proBNP, EF and Galectin-3 levels has been shown via multiple Regression Linear”
Response 4: We rewrote: Galectin-3 levels were positively correlated with age, NT-proBNP and negatively correlated between EF and galectin-3 levels
- The conclusion can not be supported by the results, which didn’t show the association between the history of coronary heart disease, NT-ProBNP levels, left ventricular systolic function, waist circumference, BMI, time to detection of heart failure and Galectin-3.
Response 5: We deleted it in new manuscript.
- Line 35-46, there are contradictions in the context. Treatment with spironolactone, the overall concentration of spironolactone reduced from 54.82 ± 26.06 to 44.20 ± 24.36 (p<0.05). However, 50mg-spironolactone vs 25mg- spironolactone, galectin-3 concentrations were 17.11 ± 20.81 vs 3.46 ± 6.81ng/ml, which means that spironolactone improved galectin-3 concentrations. In addition, in the conclusion, the authors states that spironolactone reduced galectin-3 concentrations. Please specify
Response 6: Thanks for asking us a exemplary question.
This is an important component of the study, as we demonstrated in Table 6 that galectin-3 levels decreased before and after spironolactone treatment and the magnitude of the reduction when comparing each dose of 50mg and 25mg.
So we wrote in abstract as: “After 12 weeks treatment by spironolactone, Galectin-3 concentration decreased from 54.82 ± 26.06 to 44.20 ± 24.36 (p<0.05). According to subgroup analysis, the average concentration of Galectin-3 decreased the most in the group of patients with grade 3 hypertension and NYHA class III. The 50 mg once daily dose of spironolactone significantly improved galectin-3 concentrations over the 25 mg once daily group, which was 17.11 ± 20.81 (p < 0.05) (reduced 29.05%) and 3.46 ± 6.81ng/ml (p<0.05) (reduced 6.87%), respectively”.
- Line 60, what’s the pathology of HF? Or the authors means etiology or causes?
Response 7: Sincere apologies, we should have written as galectin-3 increased in patients with heart failure and common causes of heart failure (eg coronary artery disease, hypertension...). We will revise in the manuscript.
- Line 66-67, please add references for the statement, and the use of word “current” is easy to lead to misunderstanding
Response 8: Thanks for helping us find a wording error, our research has been on the issue since 2018 so we will remove the word: "current"
- The introduction can be divided into several paragraphs to introduce the epidemiology of heart failure, characteristics of Galectin-3, characteristics of spironolactone, molecular association between Galectin-3 and spironolactone, current problems and problems that the article hopes to solve. This makes it easier for the reader to read.
Response 9: We will divide the paragraphs as comments of the reviewers.
- Line 102-105, according to the statement “Their initial dose will be 25 mg once daily, and they will be re-evaluated after 2 weeks. Patients who are symptomatic, well tolerated, and have no contraindications may have their dose increased to the target dose of 50 mg once daily”, the 50mg group were not treated with 50mg for 12 weeks but 10 weeks. Whether this should be highlighted as “25mg-2 weeks+50mg-10 weeks” group for accuracy.
Response 10: We wrote with this in reason: We initiate treatment with 25mg then increase to 50mg after 2 weeks if tolerated, no contraindications. We would estimate the total number of patients receiving the 25mg and 50mg doses after 12 weeks. As your opinion is 2 weeks 25mg and 10 weeks 50mg.
- Line 120-121, this sentence is redundant.
Response 11: We deleted it in new manuscript.
- Line 130, please delete the sentence “As a result, the data will be recorded”.
Response 12: We deleted it in new manuscript.
- Line 216, the authors state that “there was a noticeable decrease in the mean concentration of Galectin-3 in class III, from 63.04 ± 20.79 to 50.76 216± 18.70 (p<0.001)”. However, in Table 6, the decrease of Galectin-3 concentration in the NYHA class IV group is the most with P=0.05. This is because the number of patients is small (n=4). This should be noticed.
Response 13: We totally agree and add to the manuscript this argument.
- The discussion is too long. May the authors consider to make it concise
Response 14: We have condensed the non-repetitive statements to shorten the discussion.

Reviewer 3 Report
Thank you for submitting your valuable manuscript to this journal.
The aim of the present study was to determine the factors related to the high levels of galectin-3 in HFrEF patients and to evaluate the effectiveness of spironolactone on its plasma concentration.
This is a cross-sectional descriptive prospective study on 122 HFrEF patients. The methods are very well explained and the study population are defined adequately. The flow diagram of the study is well designed and illustrative. Another advantage of the study protocol is its duration of intervention which is a 3-month drug treatment to ensure its effects on the outcome measures.
There are some points that need to be considered in the paper:
1- On data collection section, it’s mentioned that the initial dose of spironolactone was 25 mg once daily and after 2 weeks it increased to 50 mg if the patients were symptomatic, well tolerated without contraindications. At the end of the study, 58 patients were still on 25 mg once daily. Please explain the reasons, disease status and risk factors of the patients who did not increase the dose of spironolactone in detail.
You can add these data for each group (25 mg vs. 50 mg) to the baseline characteristics (table 1).
2- What was the medication regimen of the patients at the beginning and during the course of the study? Please add these data in the table 1. Did they receive medications such as SGLT2 or ARNI that may affect galectin-3 levels?
3- On the results section, the baseline characteristics of the patients are repeated in the text. It is enough to show them in the table. Please remove the repeated data in the text.
4- Since the number of patients with normal galectin-3 (<22.1 ng/ml) is low in this study, it seems that it is not proper to measure odd ratio (table 2 and 5) to show the relationship of cardiovascular risk factors and galectin-3 levels. Pearson correlation is appropriate for this purpose (table 3), so please remove table 2 and 5 from the results of this study.
5- It is true that there were a linear relationship between NT-proBNP level, left ventricular systolic function (EF) and plasma Galectin-3 concentration, but estimating the level of galectin-3 is not the aim of this study, and basically it is not the case in HFrEF patients.
Therefore, please remove the formula for linear regression and table 4 form the results of the study.
6- The last row in the table 6 is for spironolactone 50 mg per day. Please correct it.
7- There are repeated data in the text at the end of the results section. It is sufficient to show the data in Table 6, so please remove repeated data from the text.
8- The discussion part of the paper is too long with unnecessary data about the basic information on galectin-3 and related factors. Please remove the redundant data and just explain the results of this study and compare them with the literature.
Good luck.
Author Response
First, the authors sincerely thank the reviewers for agreeing to read and give us valuable contributions.
Here we would like to reply to the comments of the reviewer as follows:
- On data collection section, it’s mentioned that the initial dose of spironolactone was 25 mg once daily and after 2 weeks it increased to 50 mg if the patients were symptomatic, well tolerated without contraindications. At the end of the study, 58 patients were still on 25 mg once daily. Please explain the reasons, disease status and risk factors of the patients who did not increase the dose of spironolactone in detail.
You can add these data for each group (25 mg vs. 50 mg) to the baseline characteristics (table 1).
Response 1: The group of patients using 25mg dose did not increase the dose because: patients had systolic blood pressure ranging from 100-110 mmHg during treatment. We added spironolactone value to Table 1 as a reviewer's argument.
- What was the medication regimen of the patients at the beginning and during the course of the study? Please add these data in the table 1. Did they receive medications such as SGLT2 or ARNI that may affect galectin-3 levels?
Response 2: We re-coordinated the data collected and counted the patient's background drug history preliminary to spironolactone treatment. Due to the research that we carried out from 2018 to 2019, and in Vietnam at that period ARNI and SGLT2i were not approved for use.
- On the results section, the baseline characteristics of the patients are repeated in the text. It is enough to show them in the table. Please remove the repeated data in the text.
Response 3: We removed it according to reviewer's argument
- Since the number of patients with normal galectin-3 (<22.1 ng/ml) is low in this study, it seems that it is not proper to measure odd ratio (table 2 and 5) to show the relationship of cardiovascular risk factors and galectin-3 levels. Pearson correlation is appropriate for this purpose (table 3), so please remove table 2 and 5 from the results of this study.
Response 4: We removed it according to reviewer's argument
- It is true that there were a linear relationship between NT-proBNP level, left ventricular systolic function (EF) and plasma Galectin-3 concentration, but estimating the level of galectin-3 is not the aim of this study, and basically it is not the case in HFrEF patients. Therefore, please remove the formula for linear regression and table 4 form the results of the study.
Response 5: Our study had two objectives: (i) to determine the factors related to the elevation of plasma galectin-3 concentrations in HFrEF patients and (ii) to evaluate the effectiveness of spironolactone on the modulation of plasma galectin-3 concentration. Therefore, Table 4 is necessary to represent the research objective.
- The last row in the table 6 is for spironolactone 50 mg per day. Please correct it.
Response 6: We corrected this bug
- There are repeated data in the text at the end of the results section. It is sufficient to show the data in Table 6, so please remove repeated data from the text.
Response 7: We will remove the repetitions and accentuate the main points in the results in Table 6.
- The discussion part of the paper is too long with unnecessary data about the basic information on galectin-3 and related factors. Please remove the redundant data and just explain the results of this study and compare them with the literature.
Response 8: We withdrew the parts that repeat the results, the factors that were not necessary. However, since we had mark for several factors that associate with increased galectin-3 and cardiovascular risk factors, we will keep the essential sections.

Round 2
Reviewer 2 Report
Thank you for the responses. I have no further comments.
Author Response
Thanks to the reviewer for allowing us to publish in a healthcare journal.
Reviewer 3 Report
Thank you for the revision of your valuable manuscript.
Many comments have been revised and I have no further comments.
Good luck.
Author Response

(The authors gave the same response as above.)
